# Multi-State Second-Order Nonlinear Optical Switches Incorporating One to Three Benzazolo-Oxazolidine Units: A Quantum Chemistry Investigation

**DOI:** 10.3390/molecules27092770

**Published:** 2022-04-26

**Authors:** Pierre Beaujean, Lionel Sanguinet, Vincent Rodriguez, Frédéric Castet, Benoît Champagne

**Affiliations:** 1Theoretical Chemistry Laboratory, Unit of Theoretical and Structural Physical Chemistry, Namur Institute of Structured Matter, University of Namur, B-5000 Namur, Belgium; benoit.champagne@unamur.be; 2MOLTECH-Anjou (CNRS-UMR 6200), Université d’Angers, F-49045 Angers, France; lionel.sanguinet@univ-angers.fr; 3CNRS, Bordeaux INP, ISM, Univ. Bordeaux, UMR 5255, F-33400 Talence, France; vincent.rodriguez@u-bordeaux.fr (V.R.); frederic.castet@u-bordeaux.fr (F.C.)

**Keywords:** nonlinear optics, molecular switch, second-order NLO response, NLO switch, benzazolooxazolidine

## Abstract

This contribution employs quantum chemistry methods to describe the variations of the second nonlinear optical responses of molecular switches based on benzazolo-oxazolidine (BOX) units, connected by *π*-linkers, along their successive opening/closing. Under the fully closed forms, all of them display negligible first hyperpolarizability (*β*) values. When one BOX is opened, which is sketched as **C**→**O**, a push–pull *π*-conjugated segment is formed, having the potential to enhance *β* and to set the depolarization ratio (DR) to its one-dimensional-like value (DR = 5). This is observed when only one BOX is open, either for the monoBOX species (**C**→**O**) or for the diBOX (**CC**→**CO**) and triBOX (**CCC**→**CCO**) compounds, i.e., when the remaining BOXs stay closed. The next BOX openings have much different effects. For the diBOXs, the second opening (**CO**→**OO**) is associated with a decrease of *β*, and this decrease is tuned by controlling the conformation of the *π*-linker, i.e., the centrosymmetry of the whole compound because *β* vanishes in centrosymmetric compounds. For the triBOXs, the second opening gives rise to a *Λ*-shape compound, with a negligible change of *β*, but a decrease of the DR whereas, along the third opening, *β* remains similar and the DR decreases to the typical value of octupolar systems (DR = 1.5).

## 1. Introduction

Owing to numerous application fields, molecular switches have now been studied for many years, and continue to receive attention. They are defined as molecules that adjust their structural and, therefore, their electronic, optical, etc., properties to an external stimulus, with two (or more) metastable states [1]. Such switching phenomena therefore lead to materials where one or more properties are modulated on demand. So far, a wide array of molecules has been reported where changes of color, luminescence, electrochemical potential, or other properties are the target. The nature of the trigger is generally used for their classification, resulting in photochromic (triggered by light irradiation), electrochromic (electrochemical stimulation), thermochromic (temperature), etc., compounds [1,2,3,4,5].

In particular, the field of nonlinear optical (NLO) switches has drawn attention in the past few decades, for their potential applications in optoelectronics and photonics [6,7,8,9,10]. Though third-order NLO responses can also be enacted [11,12], the NLO properties of interest are generally of second order. At the molecular level, they are described by the first hyperpolarizability (*β*) [10,13,14,15,16,17,18,19]. Experimentally, *β* is probed by techniques such as hyper-Rayleigh scattering (HRS) [20,21,22] or electric-field induced second harmonic generation (EFISHG) [23,24]. In parallel, theoretical tools have been developed to rationalize the *β* responses and improve the design of such systems [25,26,27,28].

One way to achieve systems displaying multiple states consists of combining several switching units by covalent linkage or supramolecular assembly: a system with *n* different two-state moieties can exhibit up to 2_*n*_ distinct states or *n* + 1 distinct states if the molecular system presents an intrinsic symmetry (i.e., equivalent switching units). In the past, we and others have focused on derivatives of benzazolo[2,1-b]oxazolidine (BOX) (Figure 1) [29], a two-state multiaddressable (acidochromic, photochromic, electrochromic) unit [30,31,32,33]. Thanks to an easy and straightforward synthesis route, molecular systems involving one, two (diBOX), and three (triBOX) BOX units (Figure 2) have also been studied from both the experimental [32,34,35,36,37] and quantum chemical [30,31,38,39,40] points of view. In our last contribution [41], the synthesis and characterization of unsymmetrical triBOX were performed, opening the door towards more performant systems.

By enacting the methods of density functional theory (DFT) and time-dependent DFT (TDDFT), this contribution focuses on the second-order NLO properties of multi-BOX systems (Figure 3) as a function of their level of openings. The second harmonic generation (SHG) responses that could be determined by the HRS experiment are analyzed because they provide both an amplitude (*β_HRS_*) and a parameter (the depolarization ratio (DR)) that is related to the topology of the NLOphore, as well as to the dipolar/octupolar character of the *β*-tensor [42]. One of the objectives is to reveal the variations of the NLO switching behavior when going from a compound with a single BOX unit to compounds with two or three BOXs. The reference compound bearing a single BOX unit (**1**) is similar to one compound that has been characterized by one of us a few years ago [38], though, here, R_1_ is a H atom rather than a Me group.This choice is dictated by consistency with the chemical structure of the other compounds, where R_1_ = H. Among the few diBOXs reported in the literature, the simplest has a bithiophene linker (**2a**), whereas, following [40], two of its derivatives are obtained by replacing it either with two 3,4-ethylenedioxythiopene (EDOT) units (**2d**) or with an EDOT-thiophene-EDOT sequence (**2e**). By combining experimental characterizations with (TD)DFT calculations, [40] highlighted the interplay between the molecular symmetry via controlling the dihedral angles between the aromatic rings of the linker and their linear and second-order nonlinear optical properties. This is why, to further span the range of the dihedral angle values, two other compounds are proposed here, **2b**, where the ethyl substituents disfavor planarity, and **2c**, where the cyclopentadithiophene linker is by construction planar. For consistency, all five diBOX derivatives are discussed in this work, where the same methods of calculation were employed for the whole set of compounds (this explains why for **2a**, **2d**, and **2e**, some results are quantitatively slightly different). To complete them and have a full representation of multiphotochromic systems based on BOX, two triBOX systems elaborated from triarylamine were studied as well. They differ in the sense that in **3a**, the linkers are identical (phenylthiophene units), whereas they are different (phenyl, biphenyl, and phenylthiophene) in **3b**, allowing analyzing the impact of the different sequential openings of the BOXs. For these triBOXs, the synthesis, redox, and optical properties have recently been presented and analyzed in the light of (TD)DFT calculations [41], but their second-order NLO responses have not yet been disclosed. As a matter of fact, for the whole set of compounds, the current study focuses first on their hyperpolarizability and their relation to structural, reactivity (acido-triggered opening reactions), and linear optical properties. In the present (TD)DFT contribution, each molecular state is characterized separately, whereas experimentally, several states/forms could co-exist along the successive acido- or redox-triggered switching steps. For a few compounds, HRS experimental data are available, so that comparisons with the calculations are also briefly discussed.

This paper is divided into four parts: after describing the methodological and computational methods in Section 2, the results (structural, acidochromic, and second-order NLO properties) are presented and analyzed in Section 3. Section 4 draws the conclusions.

## 2. Materials and Methods

The geometries of all compounds and of their different forms obtained by opening 1, 2, or the 3 BOXs were fully optimized at the DFT level with the *ω*B97X-D XC functional [43,44], the 6-311G(d) basis set, and by accounting for solvent effects using the integral equation formalism of the polarizable continuum model (IEF-PCM) (the solvent is acetonitrile) [45]. Real vibrational frequencies demonstrate that the optimized geometries are minima on the potential energy hyper-surface. For selected structures (open forms because they present smaller excitation energies than the fully closed ones), it has been confirmed that there is no singlet, nor triplet instabilities. Since the compounds are mostly composed of cyclic units and conjugated segments, the numbers of stable conformers in solution are rather small and the search of those conformers possessing a non-negligible weight within the Maxwell–Boltzmann (MB) statistics can be carried out in a systematic manner. This was done:by defining the key torsion angles to distinguish the main conformations,then by performing rigid scans to locate the extrema of the potential energy hyper-surface;by combining the minima of these rigid scans to preselect conformations;then by performing full geometry optimizations on the latter.

Finally, only those conformers within an energy window of 12.5 kJ mol^−1^ higher than the most stable conformer were kept to calculate the MB populations, on the basis of the Gibbs free energies, Δ*G*^0^, at 298.15 K. Such an approach is efficient to locate the key conformers because the torsion angles are far enough from each other and, in good approximation, their impact on the total energy is independent of each other, leading to a quasi-additive behavior. Furthermore, considering multiple conformers is important to evaluate the NLO responses, especially when they exhibit different symmetries. In Section 3, averaged results following the MB populations of conformers are reported. Note that it is assumed that there is no equilibrium between forms with different levels of opening when computing the MB populations.

To assess the impact of the state of opening on the structure and on the *π*-conjugation of the molecules, the bond length alternation (BLA) of the vinylic bridge between the linker and the BOX units was analyzed. Given the *π*-conjugated segment C${}_{1}$–C${}_{2}$=C${}_{3}$–C${}_{4}$, the BLA is computed as
(1)$${\mathrm{BLA}}_{1,4}=\frac{1}{2}\;\left(l_{12}+l_{34}-2\;l_{23}\right),$$
where $l_{ij}$ is the distance between carbons *i* and *j*.

Given that the linkers contain aromatic rings, steric hindrance prevents a perfect orientation of the pz orbitals, normal to the plane of the π-conjugated path. To assess this impact, given the π-conjugated segment C${}_{1}$=C${}_{2}$–C${}_{3}$=C${}_{4}$ (where C1 and C2 belong to the first aromatic cycle while C3 and C4 belong to the second), the out-of-plane angle (OOPA) is computed as
(2)$${∠}_{1,4}=\min\{|\theta_{1,4}\left|,180-\right|\theta_{1,4}\left|\right\},$$
where $|\theta_{1,4}|\in \left[0,180{}^{\circ }\right]$ is the absolute value of the dihedral angle between C_1_ and C_4_.

For each form, the NLO properties were then computed at the M06-2X/6-311+G(d) level of approximation, in acetonitrile (IEF-PCM). In a recent investigation [46], this implicit solvation approach has been challenged with respect to an explicit model where the solvent molecules are represented by point charges, of which the positions have been generated by Monte Carlo simulations, whereas the solute is treated quantum mechanically. It has been shown that both approaches predict similar contrasts, indicating that implicit solvation models such as IEF-PCM are well suited to describe the variations in the NLO responses of molecular switches. Here, we focus on the evaluation of the quantities that would be extracted from the hyper-Rayleigh scattering (HRS) experiments: $\beta_{HRS}$ and its depolarization ratio (DR) as defined by the sum and ratio of the β-tensor orientational averages [22], respectively, according to: (3)$$\begin{array}{cc} \; & \beta_{HRS}=\sqrt{\langle \beta_{ZZZ}^{2}\rangle +\langle \beta_{ZXX}^{2}\rangle }\;\mathrm{and}\;\mathrm{DR}=\frac{\langle \beta_{ZZZ}^{2}\rangle }{\langle \beta_{ZXX}^{2}\rangle }. \end{array}$$

To highlight the dipolar or octupolar structures of the NLOphores, the unit sphere representation (USR) is also given for the most stable forms, plotted using the DrawMol program [47]. In such figures, arrows represent the effective second-order induced dipoles, $\mu_{\mathrm{eff}}=\beta :E^{2}\left(\theta ,\phi \right)$, plotted at each point (*θ*,*ϕ*) of a sphere centered at the center of mass of the molecule. **E** is a unit vector of the incident electric field with polarization defined in spherical coordinates. All reported *β* values are given in a.u. (1 a.u. of *β* = 3.6212 × 10^−42^ m^4^ V^−1^ = 3.2064 × 10^−53^ C^3^m^3^J^−2^ = 8.639 × 10^−33^ esu) within the T convention [26].

Finally, to help the interpretation, linear optical properties are also computed with TDDFT. For the unpublished compounds (**1**, **2b**, and **2c**), they were computed at the M06-2X/6-311+G(d) level of approximation, in acetonitrile (IEF-PCM). For the others, they were taken from [40] (**2a**, **2d**, and **2e**) and [41] (**3a** and **3b**).

As noted, a different XC functional was used for calculating the (non)linear optical properties (M06-2X, 54% HF exchange) and the structural and thermodynamic data (*ω*B97X-D, 16% and 100% of HF exchange at the short- and long-range, respectively, with a range-separating parameter *ω* = 0.2 *a*_0_^−1^). This is consistent with previous studies on related compounds [9,40,48,49,50,51]. All (TD)DFT calculations were carried out using the Gaussian 16 package [52].

## 3. Results

### 3.1. Structural Properties

The BLA values (Table 1) were all positive. They are witnesses of the BOX opening: the BLA of the vinylidene bridge (defined in Figure 3) decreases as the corresponding BOX opens, which allows *π*-conjugation between the donor and acceptor units. The impact of the linker is also visible, since the BLA (in *Å*) in the closed form satisfies the ordering: $$\underset{0.142}{\underset{︸}{\mathrm{biE}=\mathrm{EThE}}}<\underset{0.145}{\underset{︸}{\mathrm{CpdiTh}}}<\underset{0.147-0.148}{\underset{︸}{\mathrm{biThSMe}\approx \mathrm{PhTh}\approx \mathrm{biEtTh}\approx \mathrm{biTh}}}<\underset{0.156-0.157}{\underset{︸}{\mathrm{Ph}\approx \mathrm{biPh}}}.$$

For the fully open forms (**O**, **OO**, and **OOO**), the order becomes: $$\underset{0.046}{\underset{︸}{\mathrm{EThE}}}<\underset{0.055}{\underset{︸}{\mathrm{biE}}}<\underset{0.061-0.063}{\underset{︸}{\mathrm{biThSMe}\approx \mathrm{CpdiTh}\approx \mathrm{PhTh}}}<\underset{0.070-0.073}{\underset{︸}{\mathrm{biTh}\approx \mathrm{biEtTh}\approx \mathrm{Ph}}}<\underset{0.087}{\underset{︸}{\mathrm{biPh}}}.$$

The delocalization is thus the strongest for compounds containing the EDOT fragments, followed by thiophenes, and the weakest for phenyl. A more subtle effect is evidenced by the sequential opening of di- and triBOXs: the BLA, and so the *π*-conjugation, slightly increases, revealing that the BOX units are competing. This is especially visible in **2c** (CpdiTh) and **2d** (biE), for which the BLAs of the **CO** forms are 0.2 *Å* smaller than the ones of **OO**.

The OOPA (in °) for all forms follows a similar trend: $$\underset{0-1}{\underset{︸}{\mathrm{CpdiTh}\approx \mathrm{biE}}}<\underset{3-9}{\underset{︸}{\mathrm{EThE}}}<\underset{20-30}{\underset{︸}{\mathrm{biTh}\approx \mathrm{biThSMe}\approx \mathrm{PhTh}}}<\underset{35-40}{\underset{︸}{\mathrm{biPh}}}<\underset{85-90}{\underset{︸}{\mathrm{biEtTh}}}.$$

While comparing with the trend for BLA, two deviations appear: On the one hand, as expected from the steric hindrance, biEtTh (and biPh) features large *∠* values. For **2b**, one can thus assume that there is no *π*-conjugation between the BOX units while open and that the two moieties acts independently. On the other hand, CpdiTh displays an OOPA of 0°, which is not correlated with a small BLA: the relative strength of the donor and acceptor in the structure is also important. It should also be noted that the opening tends to reduce the OOPA by 5° or less.

### 3.2. Acidochromic Properties

Using the whole set of conformers, Figure 4 shows that the opening of the BOX by protonation can be sequential, since the successive openings are less and less exergonic. This is in agreement with the experimental results [40,41]. As seen in Figure 5, the *ΔG*^0^ for the first and second opening reactions are more exergonic when (i) the BLA and/or (ii) the OOPA of the just-opened molecular moieties is smaller. In other words, when the *π*-electron delocalization is favored with the open forms, the reaction of opening is favored.

### 3.3. NLO Properties

#### 3.3.1. *β_HRS_*, Their Contrasts, and the DR

Table 2 reports the static and dynamic (*λ* = 1907, 1300 and 1064 nm) NLO properties of all compounds in their different forms. The *β_HRS_* of the fully closed form is always small and generally the smallest of all forms (except for **2d**, due to a pseudo-C_i_ symmetry of most of the conformers of the **OO** form, which leads to even smaller *β_HRS_* values at *∞* and 1907 nm than the **CC** form). Then, the behavior depends on the number of BOX units:For **1**, opening the unique BOX gives rise to a push–pull *π*-conjugated NLOphore, of which the *β* response is much larger (from one to two orders of magnitude) than for the closed form;For diBOX (**2a**–**2e**), the order is **CC** < **OO** < **CO** with, usually, large contrasts for the first opening reaction (Figure 6), while the second contrast depends much on the *π*-linker;For triBOX (**3a**, **3b**), the *β_HRS_* of the open forms (**CCO**, **COO**, **OOO**) are similar, resulting in contrasts close to 1 for the second and third openings.

Moreover, there is generally a large enhancement of *β_HRS_* at 1064 nm for the open forms, which indicates (near) resonance with a low-lying dipole-allowed excited state.

For the diBOXs in their **CO** form, the lowest *β_HRS_* is found for the biEtTh linker (**2b**) due to the steric hindrance between the thiophene substituents, leading to a large dihedral angle and a reduced *π*-electron delocalization. Then, all the other **CO** diBOXs display a larger *β_HRS_* than **1O**, the largest being achieved by the EThE linker (**2e**). The ordering of the *β_HRS_* values is first driven by *π*-conjugation as measured by small dihedral angles and small BLAs. This explains the following ordering:
**2b** < **1** < **2a** < **2c** ≈ **2d** < **2e**.

However, this does not necessarily translate into the largest contrasts: for the first opening reaction (**CC** →**CO**), it is **2d**, followed by **2a** and then by **2e**, which displays contrasts equivalent to or larger than compound **1**. For the second opening reaction (**CO**→**OO**), the contrasts are computed using **OO** as the reference: *β_HRS_*(**CO**)/*β_HRS_*(**OO**). They are, again, large for **2a**, **2d**, and **2e**. It should be noted, however, that all contrasts (including the one between **CC** and **OO** forms, blue arrows in Figure 6) should be large to distinguish the three forms, which is not the case for **2d**. Finally, the DRs are generally large and close to 5 (i.e., typical for 1-D NLOphores) for **CO**, while close to 3 for the **CC** and **OO** forms (i.e., typical for *Λ*-shaped structures [53]). The exception is **2e**, for which the **CC** forms include non-centrosymmetric conformers with non-negligible MB weights (Appendix A).

Turning to the triBOX compounds, Table 2 reports the MB averaged values, while Table 3 details, for compound **3b**, the impact of the different linkers bearing open or closed BOXs: for **CCO**, *β_HRS_* follows the biPh < Ph < PhTh ordering, where the designated linker is attached to the open BOX. For **COO**, the largest value is achieved when the open BOXs are linked to the Ph and PhTh units. Those two observations are consistent with the BLA values (the smaller the BLA, the larger the *β_HRS_* value). Then, the evolution of the DR is prototypical: at first opening, the triBOX compound goes from octupolar (DR ~ 1.7, due to a C_3_-like topology) to linear (DR ~ 5), since the response of the latter is dominated by a single BOX unit, with one preferential charge-transfer direction. Then, the octupolar character increases with the second opening (i.e., typical for *Λ*-shaped compounds [53]). Finally, when forming **OOO**, the octupolar character is restored for **3a**, though less marked for **3b**, owing to the non-equivalence of the linkers. This latter issues is especially visible with biPh, which breaks the *π*-conjugation with the rest of the structure. Nevertheless, the DR provides an (additional) way to differentiate between the different forms.

#### 3.3.2. Unit Sphere Representations

The change of DR for the different forms is illustrated by the USRs (Figure 7 and Appendix A). One can easily distinguish between (i) the dipolar NLOphores, where the induced dipoles are oriented along the push–pull *π*-conjugation axis, from the acceptor towards the donor group (e.g., Figure 7A), (ii) the *Λ*-shaped NLOphores having two dominant *β* components, *β_zzz_* and *β_zyy_*, with *z* parallel to the *C*_2_-axis (Figure 7B), and (iii) the octupolar compounds with *β_yyy_* = −*β_yxx_*, with *y* one of the *C*_2_ axes (Figure 7C), for perfect octupolar and planar molecules (D_3h_). In the case of the *Λ*-shaped compounds, the vector field is characterized by dominant induced dipoles that form a cross (letter x) with non-orthogonal branches. Then, in octupolar systems, they define three directions, equidistant when the three linkers are identical (**3a**), while slightly distorted when they are different (**3b**).

#### 3.3.3. Comparison with Experiments

Comparison with an experiment gives the opportunity to assess the reliability of our methodology. On the one hand, experimental *β_HRS_* have been reported for compounds **1** (with R = Me and NO_2_) [38] and **2a** [40]. In both cases, the TDDFT M06-2X approach reproduces the experimental trends, and the calculation gives responses of the same order of magnitude as the measured results, though care should be used when dealing with resonance. On the other hand, when comparing the predicted and experimental lowest excited-state energy, which contributes substantially to the UV/VIS absorption spectrum (Appendix A), there is a systematic overestimation (by about 0.3 eV) of the excitation energies, which can be explained by the fact that the calculated values are vertical excitation energies, while the experimental ones are the maxima of absorption.

#### 3.3.4. Further Analysis

Few-state analyses have often been used to interpret the NLO responses. For dipolar systems, the dominant *β_zzz_* component is expressed within the two-state approximation [54,55,56], involving one ground (labeled *g*) and one excited (labeled *e*) state. In the static limit, it reads:(4)$$\beta_{HRS}=\sqrt{\frac{6}{35}}\;\beta_{zzz}=6\;\sqrt{\frac{6}{35}}\;\frac{\Delta \mu_{ge}\;\mu_{ge}^{2}}{\Delta E_{ge}^{2}},\;\mathrm{with}\;\Delta \mu_{ge}=\mu_{e}-\mu_{g},$$
where $\Delta E_{ge}$ is the excitation energy, $\mu_{ge}$ is the transition dipole moment, and $\Delta \mu_{ge}$ is the difference between the ground and excited state dipole moments. For *Λ*-shaped and octupolar compounds, at least two excited states (*e* and *e*′) should be considered so that the dominant *β*-tensor components are proportional to $\Delta E_{ge}^{-2}$ and $\Delta E_{ge}^{-1}\Delta E_{ge^{\prime }}^{-1}$. In a first approximation, Figure 8 tackles the possible relationship between *β_HRS_* and $\Delta E_{ge}^{-2}$. Here, $\Delta E_{ge}$ is the excitation energy of the lowest-energy dipole-allowed electronic transition. Note that for the *Λ*-shaped and octupolar compounds, the second excitation energy $\Delta E_{ge^{\prime }}$ is similar to $\Delta E_{ge}$. Figure 8 clearly distinguishes between (i) the fully closed forms with negligible *β_HRS_* responses, (ii) the quasi-symmetric **OO** forms (**2a**, **2b** and **2d**) with small *β_HRS_* values, (iii) the other **OO** forms that adopt a *Λ*-shaped structure (**2c** and **2e**), and (iv) the other compounds that present similar $\beta_{HRS}-\Delta E_{ge}$ relationships.

## 4. Conclusions and Outlooks

In this study, molecular switches containing one to three BOX units were studied using quantum chemistry calculations carried out at the DFT and TDDFT levels. Different structures were considered, and the number of states depends on: (i) the number of BOXs (**1**, monoBOX; **2**, diBOX; **3**, triBOX) and (ii) the nature of the *π*-conjugated linker (**3a** versus **3b**). Calculations showed that:The first opening leads to a drastic change of the NLO responses (at most, a tenfold increase of *β_HRS_* accompanied by an increase of the DR), driven by an enhancement of the push–pull *π*-conjugation.The following openings see either a decrease (diBOXs) or a modest variation (triBOX) of *β_HRS_*.Nevertheless, these second (and third) openings are also accompanied by a change of the depolarization ratio, which may help to differentiate between the forms.The opening mechanism upon protonation is sequential, and the trend of exergonicity is also in phase with the *π*-conjugation.

These results were rationalized by using unit sphere representations, revealing the symmetry of the *β*-tensor, and the few-state approximation. Since the nature and contrast of the *β* responses for the different forms depend on the linker, improving the design of the triBOX, in order to better differentiate the *β_HRS_* between the different forms, is an option. An interesting tool to rationalize the results for such multi-BOX compounds is the VB-*n*CT model (with *n* = 2 for diBOX [53] and *n* = 3 for triBOX [57]), as done recently for ruthenium-based NLO switches [58]. Other schemes, such as field-induced [59] or natural transition [60] orbitals, could also be considered.

## Figures and Tables

**Figure 1 molecules-27-02770-f001:**
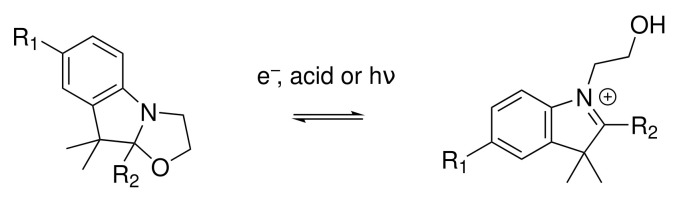
The benzozalooxazoline (BOX) multi-addressable NLO switch (see, e.g., [33]). The form of the left- (right-) hand side is referred to as the “closed” (“open”) form, displaying the smallest (largest) *β* value. To enhance the response of the open form, R_1_ is generally an acceptor group (or it is the grafting point in the case of multi-state switches), while R_2_ is ideally a donor.

**Figure 2 molecules-27-02770-f002:**
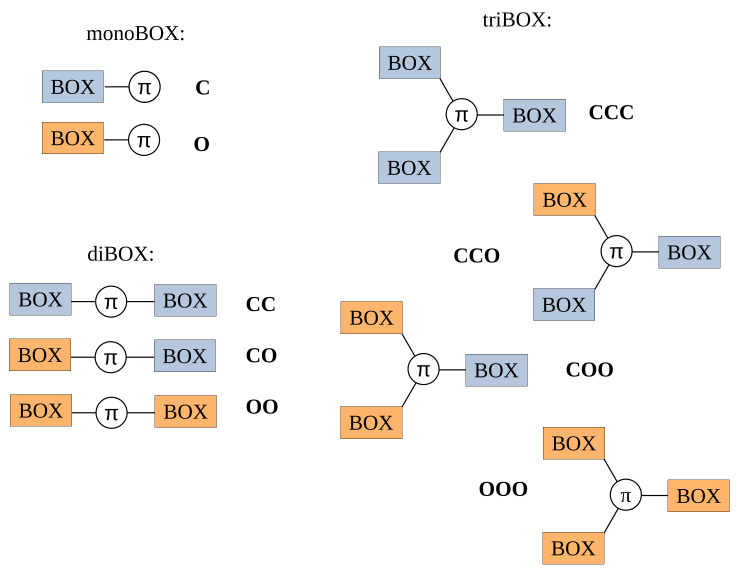
Schematic representation of the different forms associated with a monoBOX, diBOX, or triBOX, as a function of the closed (**C**) or open (**O**) state of each BOX unit.

**Figure 3 molecules-27-02770-f003:**
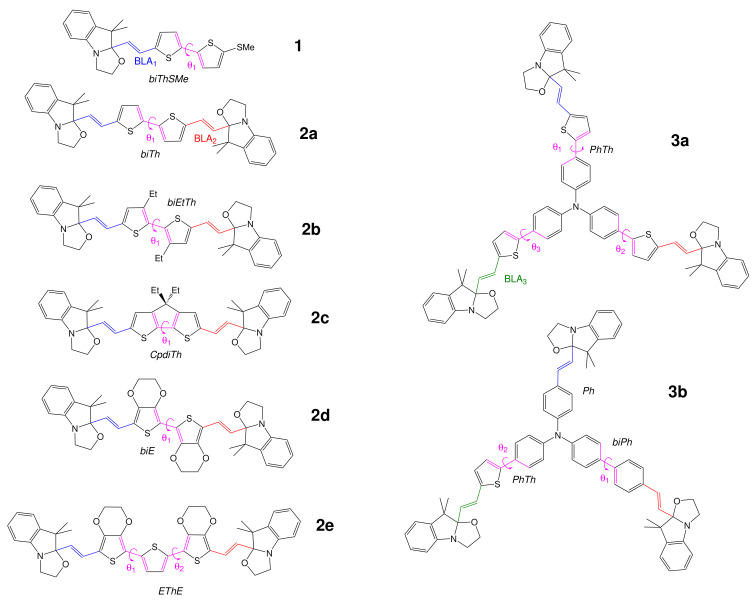
Structure of the different BOX derivatives, with the nomenclature of their *π*-conjugated linkers (bithiophene (biTh), biethylthiophene (biEtTh), cyclopentadithiophene (CpdiTh), bi-EDOT (biE), EDOT-thiophene-EDOT (EThE), phenyl (Ph), biphenyl (biPh), and phenylthiophene (PhTh)), as well as the definition of the different segments where the BLA and out-of-plane angles (OOPAs) are calculated.

**Figure 4 molecules-27-02770-f004:**
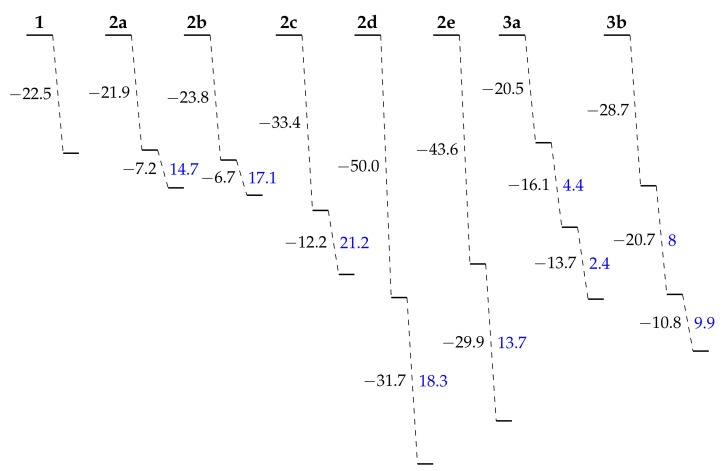
Average Gibbs free energy ($\Delta G_{i}^{0},i\in \left[1,3\right]$, kJ mol^−1^, computed as the sum of the energies of the independent fragments) at 298.15 K of the successive protonation reactions with trifluoroacetic acid (TFA) as a function of the level of opening. For the second (and third) openings, $\Delta \Delta G^{0}=\Delta G_{i}^{0}-\Delta G_{i-1}^{0}$ (kJ mol^−1^), being the difference between the current and previous $\Delta G^{0}$, is also reported in blue. The calculations used the *ω*B97X-D/6-311G(d)/IEF-PCM (acetonitrile) level of theory for the BOXs and the *ω*B97X-D/6-311+G(d)/IEF-PCM (acetonitrile) level of theory for the TFA. For **3b**, only the most spontaneous sequence of opening (Ph → PhTh → BiPh) is reported.

**Figure 5 molecules-27-02770-f005:**
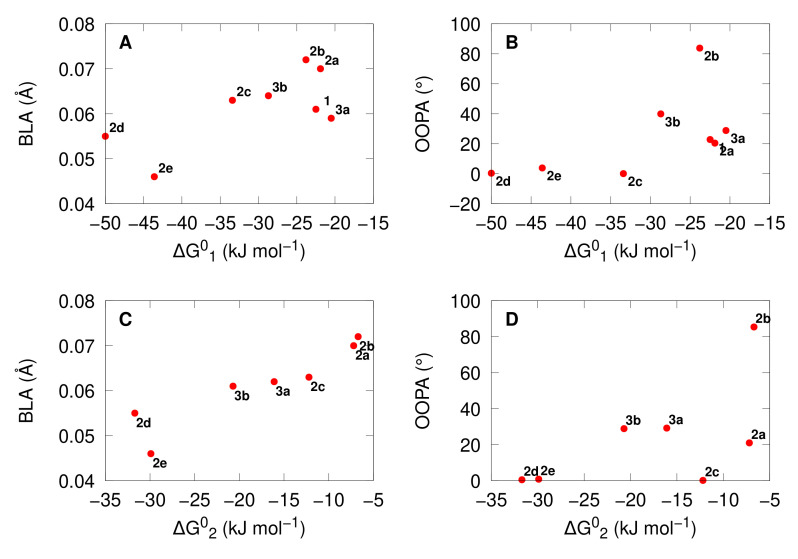
Correlation between the BLA (*Å*, panels **A** and **C**) and OOPA (°, panels **B** and **D**) of the open moiety with the corresponding Gibbs free energy along the first ($\Delta G_{1}^{0}$, kJ mol^−1^, panels **A** and **B**) and second ($\Delta G_{2}^{0}$, kJ mol^−1^, panels **C** and **D**) protonation reactions with TFA. The calculations used the *ω*B97X-D/6-311G(d)/IEF-PCM (acetonitrile) level of theory for the BOXs and the *ω*B97X-D/6-311+G(d)/IEF-PCM (acetonitrile) level of theory for the TFA.

**Figure 6 molecules-27-02770-f006:**
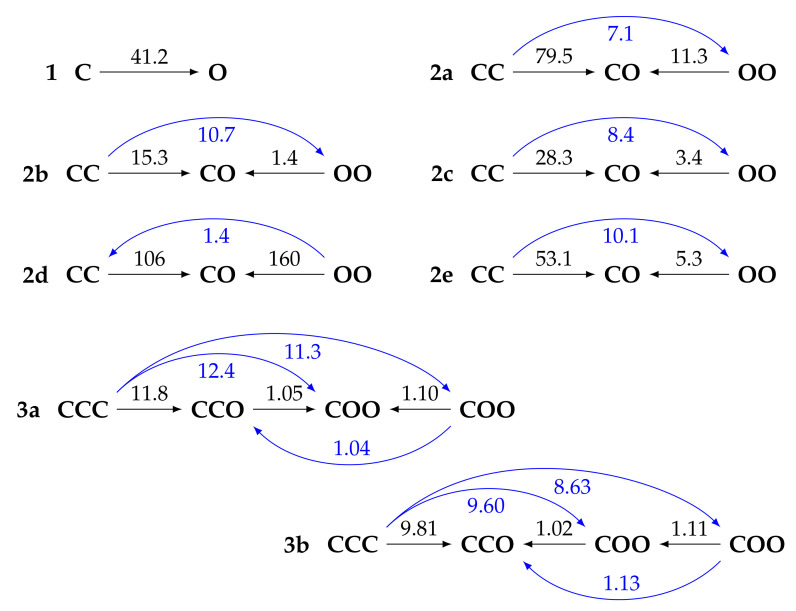
Contrasts of dynamic (at 1907 nm) first hyperpolarizability (given on the arrows by *β*(end)/*β*(start)) of the compounds in their different forms, as evaluated at the TDDFT/M06-2X/6-311+G(d)/IEF-PCM (acetonitrile) level of approximation. The blue arrows are obtained by transitivity. These are averaged values using the MB populations at 298.15 K as calculated at the *ω*B97X-D/6-311G(d)/IEF-PCM (acetonitrile) level of theory.

**Figure 7 molecules-27-02770-f007:**
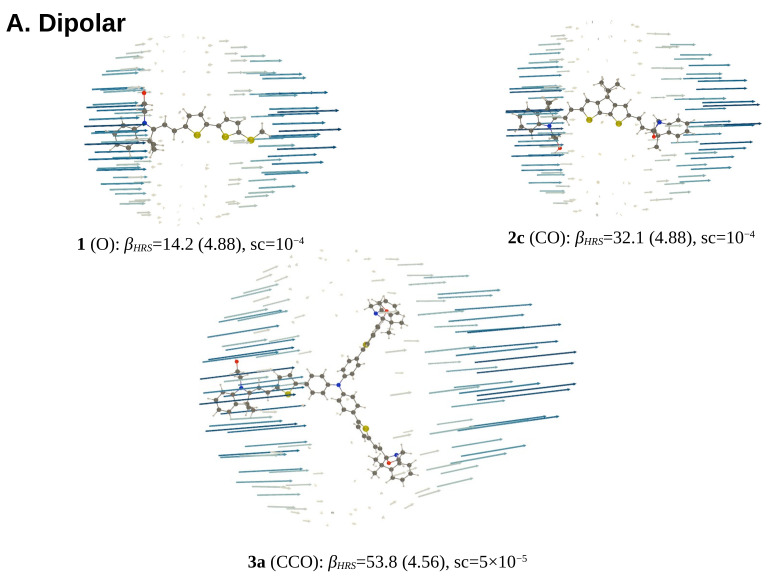
USR (together with *β_HRS_* (in 10^3^ a.u., the DR in parentheses) and the scaling factor (sc, *Å* a.u. ^−1^)) of the dynamic (*λ* = 1907 nm) *β*-tensor of the most stable conformers of selected dipolar (panel A), *Λ*-shaped (panel B) and octupolar (panel C) NLOphores, as evaluated at the TDDFT/M06-2X/6-311+G(d)/IEF-PCM (acetonitrile) level of approximation.

**Figure 8 molecules-27-02770-f008:**
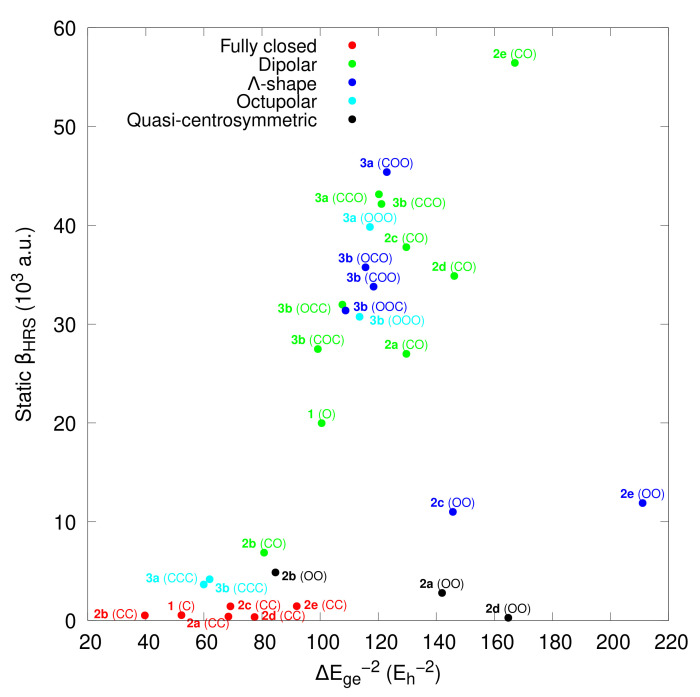
Correlation between the static *β_HRS_* and $\Delta E_{ge}^{-2}$ for the different types of NLOphores. All values were calculated at the TDDFT/M06-2X/6-311+G(d)/IEF-PCM (acetonitrile) level of approximation.

**Table 1 molecules-27-02770-t001:** BLA values (*Å*) and out-of-plane angles (*∠*, °, computed from the dihedral angles *θ*) of the different forms of the compounds, as defined in Figure 3 and evaluated at the *ω*B97X-D/6-311-G(d)/IEF-PCM (acetonitrile) level of theory. These are averaged values using the MB populations at 298.15 K as calculated at the *ω*B97X-D/6-311G(d)/IEF-PCM (acetonitrile) level of theory.

	Form	BLA_1_	BLA_2_	BLA_3_	${∠}_{1}$	${∠}_{2}$	${∠}_{3}$
**1** (biThSMe)	**C**	0.148	—	—	29.5	—	—
	**O**	0.061	—	—	22.8	—	—
**2a** (biTh)	**CC**	0.147	0.147	—	25.5	—	—
	**CO**	0.148	0.060	—	20.4	—	—
	**OO**	0.071	0.070	—	21.0	—	—
**2b** (biEtTh)	**CC**	0.148	0.148	—	88.9	—	—
	**CO**	0.148	0.068	—	83.8	—	—
	**OO**	0.073	0.072	—	85.4	—	—
**2c** (Cpdith)	**CC**	0.145	0.145	—	0.05	—	—
	**CO**	0.147	0.042	—	0.04	—	—
	**OO**	0.063	0.063	—	0.15	—	—
**2d** (biE)	**CC**	0.142	0.142	—	0.99	—	—
	**CO**	0.143	0.037	—	0.36	—	—
	**OO**	0.055	0.055	—	0.51	—	—
**2e** (EThE)	**CC**	0.142	0.142	—	3.3	8.8	—
	**CO**	0.143	0.042	—	3.8	2.3	—
	**OO**	0.046	0.046	—	0.8	5.3	—
**3a** (PhTh)	**CCC**	0.147	0.147	0.147	31.2	30.7	31.3
	**CCO**	0.147	0.147	0.059	31.9	30.5	28.8
	**COO**	0.147	0.061	0.061	31.2	29.2	28.7
	**OOO**	0.062	0.062	0.062	28.9	30.2	29.7
**3b**	**Ph(C)-BiPh(C)-PhTh(C)**	0.156	0.157	0.147	39.4	31.7	—
	**Ph(C)-BiPh(C)-PhTh(O)**	0.157	0.157	0.056	39.9	26.6	—
	**Ph(C)-BiPh(O)-PhTh(C)**	0.156	0.085	0.147	37.4	31.2	—
	**Ph(O)-BiPh(C)-PhTh(C)**	0.064	0.157	0.147	39.9	31.0	—
	**Ph(C)-BiPh(O)-PhTh(O)**	0.157	0.086	0.058	39.2	28.9	—
	**Ph(O)-BiPh(C)-PhTh(O)**	0.069	0.157	0.061	39.8	31.0	—
	**Ph(O)-BiPh(O)-PhTh(C)**	0.067	0.087	0.147	35.7	28.6	—
	**Ph(O)-BiPh(O)-PhTh(O)**	0.070	0.087	0.062	39.2	29.6	—

**Table 2 molecules-27-02770-t002:** Static and dynamic (1907, 1300, and 1064 nm) first hyperpolarizabilities (*β_HRS_* in 10^3^ a.u., the DR in parentheses) of compounds **1**–**3** in their different forms, as evaluated at the TDDFT/M06-2X/6-311+G(d)/IEF-PCM (acetonitrile) level of approximation. These are averaged values using the MB populations at 298.15 K as calculated at the *ω*B97X-D/6-311G(d)/IEF-PCM (acetonitrile) level of theory.

	Form	Static	1907 nm	1300 nm	1064 nm
**1**	**C**	0.5 (3.78)	0.4 (3.67)	0.5 (3.71)	0.6 (3.71)
	**O**	20.0 (4.82)	16.7 (4.89)	29.9 (4.94)	77.3 (4.97)
**2a**	**CC**	0.4 (4.01)	0.3 (3.86)	0.4 (3.91)	0.5 (3.96)
	**CO**	27.0 (4.81)	26.1 (4.90)	50.8 (4.95)	148.5 (4.98)
	**OO**	2.8 (2.33)	2.3 (2.43)	4.2 (2.57)	14.4 (2.31)
**2b**	**CC**	0.5 (5.11)	0.4 (3.99)	0.4 (4.04)	0.5 (4.07)
	**CO**	6.9 (4.48)	6.1 (4.61)	8.9 (4.74)	14.4 (4.87)
	**OO**	4.9 (2.43)	4.2 (2.47)	6.1 (2.54)	9.5 (2.62)
**2c**	**CC**	1.4 (3.30)	1.2 (3.08)	1.5 (3.12)	1.9 (3.13)
	**CO**	37.8 (4.79)	32.9 (4.88)	74.6 (4.96)	666.6 (5.02)
	**OO**	11.0 (2.44)	9.7 (2.58)	22.6 (2.54)	92.5 (1.09)
**2d**	**CC**	0.4 (4.02)	0.3 (3.79)	0.4 (3.95)	0.5 (4.12)
	**CO**	34.9 (4.83)	32.1 (4.91)	75.7 (4.96)	1009.0 (4.94)
	**OO**	0.3 (3.75)	0.2 (3.23)	0.5 (2.82)	2.7 (2.85)
**2e**	**CC**	1.4 (5.15)	1.1 (5.09)	1.6 (5.07)	2.5 (4.62)
	**CO**	56.4 (4.92)	61.1 (4.97)	174.2 (4.98)	1523.2 (4.93)
	**OO**	11.9 (2.66)	11.6 (2.65)	33.7 (2.72)	55.5 (0.20)
**3a**	**CCC**	4.2 (1.72)	4.4 (1.64)	6.0 (1.61)	8.6 (1.59)
	**CCO**	43.1 (4.47)	52.5 (4.54)	114.9 (4.74)	528.1 (4.96)
	**COO**	45.4 (2.92)	55.2 (2.91)	116.2 (3.07)	514.4 (2.94)
	**OOO**	39.8 (1.52)	50.3 (1.47)	105.7 (1.46)	410.4 (1.46)
**3b**	**CCC**	3.6 (1.64)	3.8 (1.63)	5.0 (1.65)	7.0 (1.68)
	**CCO**	35.6 (4.65)	37.3 (4.70)	76.7 (4.82)	281.9 (4.93)
	**COO**	35.2 (3.22)	36.5 (3.15)	68.8 (3.28)	203.5 (3.20)
	**OOO**	30.7 (3.24)	32.8 (3.16)	62.4 (3.37)	179.2 (3.84)

**Table 3 molecules-27-02770-t003:** Details of the static and dynamic first hyperpolarizabilities (*β_HRS_* in 10^3^ a.u., the DR in parentheses) of triBOX **3b** in their different forms after one or two protonations (and BOX openings), as evaluated at the TDDFT/M06-2X/6-311+G(d)/IEF-PCM (acetonitrile) level of approximation. These are averaged values using the MB populations at 298.15 K as calculated at the *ω*B97X-D/6-311G(d)/IEF-PCM (acetonitrile) level of theory.

Form	Static	1907 nm	1300 nm	1064 nm
**Ph(C)-BiPh(C)-PhTh(O)**	42.2 (4.66)	48.3 (4.69)	108.8 (4.83)	502.4 (4.92)
**Ph(C)-BiPh(O)-PhTh(C)**	27.5 (4.13)	33.5 (4.31)	62.5 (4.52)	148.9 (4.75)
**Ph(O)-BiPh(C)-PhTh(C)**	32.0 (4.67)	30.8 (4.71)	58.0 (4.82)	154.6 (4.93)
**Ph(C)-BiPh(O)-PhTh(O)**	33.8 (2.93)	39.3 (2.92)	83.6 (3.28)	347.8 (3.98)
**Ph(O)-BiPh(C)-PhTh(O)**	35.8 (3.23)	36.9 (3.16)	69.8 (3.27)	206.6 (3.11)
**Ph(O)-BiPh(O)-PhTh(C)**	31.4 (3.25)	32.7 (3.16)	58.2 (3.38)	151.3 (3.85)

## Data Availability

Not applicable.

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
