# Peer review of "Multi-State Second-Order Nonlinear Optical Switches Incorporating One to Three Benzazolo-Oxazolidine Units: A Quantum Chemistry Investigation"

_molecules, 2022, doi:10.3390/molecules27092770_

Round 1
Reviewer 1 Report
Authors employed electronic-structure methods to perform in-depth analysis of second-order nonlinear optical properties of molecular switches based on benzazolo-oxazolidine moieties. These properties were studied upon successive opening/closing of the moieties in question. The methodology was validated by experimental data,
what further strengthens the results of analyses based on computer
simulations. The topic is interested to broad readership, the manuscript presents new insights into "structure-NLO property" relations for this family of switches. Moreover, the manuscript conforms to high standards of academic writing. I recommend publication after minor revision (another review is not needed).
p.1 : oxazoldine -> oxazolidine
p.5 : A reference to T-convention should be provided.
p.5 : Authors may want to consider the following reference [DOI: 10.3390/molecules26247434]
which supports their choice of one of Minnesota functionals for studies of
optical properties. In that work it was demonstrated that modern Minnesota functionals work reasonably well in predictions of excited-state density changes which link that work with second-order NLO properties.
p.7 : /caption of Fig. 4/ free Gibbs energy -> Gibbs free energy (the latter terms is in common use); see also description of supplementary materials.
p.12: In secion 3.3.4 Authors may want to provide reference to generalized
few-state model for 1st hyperpolarizability [DOI: 10.1063/5.0010231] which was developed recently.
Reviewer 2 Report
The manuscript presented by Beaujean, Champagne et al, entitled “Multi-State Second-Order Nonlinear Optical Switches Incorporating from One to Three Benzazolo-Oxazolidine Units: a Quantum Chemistry Investigation" is a very interesting work about switchers based on incorporating from one to three units of benzazolo-oxazolidine (a molecular switch formed by one 6-benzene and two 5-atom rings). One of these rings can be opened through light, an electron transference or acid medium., and they are bonded by pi linkers. These systems can experiment high nonlinear optical responses, analyzed by means of the first hyperpolarizability (beta), when the aforementioned ring is opened. This is a computational work but strongly referring experimental results of other works when the molecules are synthetically available.
The manuscript is correctly written, well structured and, in my opinion, easy to read for researchers out of the topic. As mentioned, several structures analyzed in this work (1 and 2a, for example) were compared with the experimental counterparts, reproducing the experimental trends.
I recommend a minor revision due to the fact I have found some "typos" and some points I would like the authors discuss in more detail.
With respect to the small errors in the text I found:
* Line 22: a wide array [...] "have" been reported.
* Line 69: "multiphotochormic".
* Line 154: "biThEt", but in the rest of the manuscript they use "biEtTh".
In relation to the "Materials and Methods", in line 89 the real vibrational frequencies are not able to demonstrate that the optimized geometries are minima without a stability calculation of the wavefunction (https://gaussian.com/stable/). I cite the text from the Gaussian web page: "Note that analytic frequency calculations are only valid if the wavefunction has no internal instabilities." I would like to know if the authors verified this in their calculations.
In the part of the solvent model I think the authors have decided a good option (IEF-PCM) according to the equilibrium between computational cost and quality of the results, but other researchers using QM/MM models could criticize the quality of the results in the future. That is to say, the authors should remark the good points of this model in the Materials and Methods section, the good reproduction of the existent experimental results, and, briefly, discuss if they would predict important differences adding explicit solvent molecules and using QM/MM models.
The most controversial part of the manuscript is the use of the geometrical parameters. The authors are not thorough relating these with delocalization and steric hindrance. Direct quantum observables do not exist to obtain the latter properties, and lots of works have been published in high-impact journals about computational chemistry about how to explain them, defining the observable properties with a better agreement in comparison with geometrical parameters. For example, if the authors need to analyze the delocalization, instead of using the BLA parameter, some 2-delocalization indices could be carried out (using Multiwfn). In the case of steric hindrance and hyperconjugation, the works of Jenkins et al seem to be promising by using the NG-QTAIM. However, I think this is out of the scope of the work, and I recommend to remove any reference about delocalization and steric hindrance from the text.
In the case of the "Further analysis" section, in my opinion a sum-over-states (very computationally expensive) or a field-induced orbitals study (DOI: 10.1039/c8cp07362g ) should let a better analysis, and more information about the dominant electronic transitions.
Reviewer 3 Report
The authors report a very interesting computational study as far as concerns the design and the theoretical investigation of the NLO character (second order) and its tune. Design of multi-state molecular switches is of high interest with potential application in the field of molecular electronics and the adjustment of the interaction of light with matter.
The whole computational protocol is of high quality, accurate, consistent allowing to capture the concept of the study.
- I have one issue which the authors could find of interest to comment. It is known that in photochromic multi-state molecular systems (CC,CO,OO, CCO...) not all of the states are experimentally accessible. Excitation energy transfer from the donor to the acceptor may affect the number of distinct molecular states that are experimentally accessible. Possible paremeters that could affect energy transfer are the solvent, the π- molecular skeleton, intermolecular interactions etc. My question is whether that authors can comment on the possible existence of the various molecular states, as shown in table 1.
- How HOMO-LUMO is altered among the various states ?
Minor issue
line 20, the dots after optical should be deleted,
line 168 contrats should be contrast (?)
For table S20 provide the Maxwell-Boltzmann equation employed for the computation of the averaged properties (could useful for the non-familiar readers)
